# Association between Dental Scores and Saliva Uremic Toxins

**DOI:** 10.3390/toxins15110666

**Published:** 2023-11-20

**Authors:** Claire Rigothier, Sylvain Catros, Antoine Bénard, Johan Samot, Olivier Quintin, Christian Combe, Islam Larabi, Ziad Massy, Jean-Claude Alvarez

**Affiliations:** 1Service de Néphrologie-Transplantation-Dialyse-Aphérèses, Hôpital Pellegrin, CHU de Bordeaux, 33076 Bordeaux, France; christian.combe@chu-bordeaux.fr; 2INSERM, Bio-Ingénierie Tissulaire (BioTis), UMR 1026, Université de Bordeaux, 33076 Bordeaux, France; sylvain.catros@u-bordeaux.fr; 3Département de Chirurgie Orale, CHU de Bordeaux, 33076 Bordeaux, France; johan.samot@u-bordeaux.fr; 4Clinical Epidemiology Unit (USMR), CHU de Bordeaux, 33076 Bordeaux, France; antoine.benard@u-bordeaux.fr (A.B.); o.quintin@qmul.ac.uk (O.Q.); 5Service de Pharmacologie-Toxicologie, Hôpital Raymond Poincaré, AP-HP, 92380 Garchesy, France; islamamine.larabi@aphp.fr (I.L.); jean-claude.alvarez@aphp.fr (J.-C.A.); 6MasSpecLab, Plateforme de Spectrométrie de Masse, Inserm U-1173, Université Paris Saclay (Versailles Saint Quentin-en-Yvelines), 78035 Garches, France; 7Centre for Research in Epidemiology and Population Health, Inserm UMRS 1018, Clinical Epidemiology Team, Université Paris Saclay (Versailles Saint Quentin-en-Yvelines), 78035 Villejuif, France; ziad.massy@aphp.fr; 8Department of Nephrology, CHU Ambroise Paré, APHP, Boulogne Billancourt, 92100 Paris Cedex, France

**Keywords:** dental health, chronic kidney disease, uremic toxins, saliva

## Abstract

Dental health is frequently altered in patients with chronic kidney disease. We conducted a prospective study on dental health in CKD patients with a specific interest in the association between dental health issues and the accumulation of uremic toxins in the saliva. A total of 88 patients were included in the study, with chronic kidney disease stage 2 to 5 (without kidney replacement). We analysed the total concentrations of eight uremic toxins (trimethylamine N-oxide -TMAO-, Indoxyl Sulfate, P-cresyl-sulfate, Indole 3-acetic acid, 3-carboxy-4-methyl-5-propyl-2-furanpropanoic acid -CMPF-, Kynurenine, Hippuric acid and Phenylacetylglutamine) and three precursors of uremic toxins (Tyrosine, Phenylalanine and Tryptophan) in the saliva using LC-MS/MS. We observed, for the first time, the association between various dental scores: DMFT, FST, CPITN, and OHIS, and saliva uremic toxins and precursors: TMAO, indoxyl sulfate, or hippuric acid. Further prospective interventional studies are required to confirm our results.

## 1. Introduction

Dental health is frequently altered in patients with chronic kidney disease (CKD), with consequences such as anorexia, stomatitis, or chronic inflammation. In CKD patients, periodontitis, halitosis, and the modification of dental pulp or enamel have been reported. Several studies have explored the interrelationship between oral health and CKD [1]. Most of these studies have been focused on haemodialysis patients or end-stage renal disease. A recent cross-sectional study conducted in haemodialysis patients reported an association between periodontitis and malnutrition–inflammation–atherosclerosis [2]. Thus, dental status in CKD patients might directly influence their morbi-mortality, with a predominant impact on the cardiovascular system.

The deterioration of oral health may be linked to age, poor local hygiene, socio-educational level, diabetes, a high sugar diet, tobacco smoking, xerostomia, stress, and the presence of inappropriate oral prostheses. Specific CKD factors have been shown to influence dental health: the consequences of uremic syndrome and its endocrine, immunological, and therapeutic complications [3]; abnormal mineral homeostasis with hyperphosphatemia or/and the elevation of the parathyroid hormone; and bone disorders with altered jaw bones. However, the role of uremic toxins is still unknown.

We hypothesised that these CKD dental health abnormalities might be associated with the accumulation of uremic toxins in the saliva of CKD patients.

## 2. Results

Eighty-eight CKD patients were included in the analysis. Forty-eight (54.5%) patients were males with an age range between 35 and 80. The mean (standard derivation) body mass index (BMI) was 28.5 kg/m^2^ (17.58–45.31). Thirty-six (41.9%) patients reported no tobacco smoking, and 42 (49.4%) patients reported no alcohol consumption. The distribution of CKD stage was as follows: 23 (26.4%) stage 2, 34 (39.1%) stage 3, 23 (26.4%) stage 4, and 7 (8.1%) stage 5. Diabetic nephropathy represented 18% of our study population. No fasting status was required of the patients.

The total concentrations of eight uremic toxins (trimethylamine N-oxide -TMAO-, Indoxyl Sulfate, P-cresyl-sulfate, Indole 3-acetic acid, 3-carboxy-4-methyl-5-propyl-2-furanpropanoic acid -CMPF-, Kynurenine, Hippuric acid, and Phenylacetylglutamine), and three precursors of uremic toxins (Tyrosine, Phenylalanine, and Tryptophan) in the saliva were evaluated via the method of liquid chromatography coupled to tandem mass spectrometry (LC-MS/MS). LC-MS/MS is a method that allows us to identify and/or quantify substances, combining liquid chromatography and mass spectrometry.

The levels of uremic toxin and precursors detected in the saliva of CKD patients are presented in Table 1 and Appendix A. 

We then analysed the association between uremic toxins and precursors in the saliva and four scores focused on dental health and diseases: Decayed Missing Filled per Tooth (DMFT), Filled and Sound Teeth (FST), Community Periodontal Index of Treatment Needs (CPITN), and Simplified Oral Hygiene Index (OHIS).

We observed a significant association between a worsening of the DMFT score and the detection in the saliva of hippuric acid or phenylacetylglutamine (*p* = 0.005 and 0.046, respectively); a decrease in FST score and the detection of hippuric acid (*p* = 0.02); and an improvement in the CPITN score and an increase in saliva trimethylamine N-oxide (TMAO, *p* = 0.04), using univariate analysis (Appendix A).

The results of the multivariate analyses are presented in Table 2. 

In multivariate analysis, the DMFT score worsened significantly with the age and BMI of patients for all uremic toxins and precursors (Appendix A). 

We observed an association between FST scores and the saliva levels of TMAO, hippuric acid, phenylacetylglutamine, indoxyl sulfate, and Indol-3-acetic acid. The FST (functional teeth) score decreased with an increase in TMAO (per 10 units, β = 0.02, *p* = 0.05) and the detection of hippuric acid (β = 0.57, *p* = 0.018). 

Dental hygiene (OHIS score) alteration was associated with phenylacetylglutamine detection in saliva (β = −0.29, *p* = 0.03) or an increase in p-cresyl-sulfate (per 100 units, β = −0.05, *p* = 0.11).

We also observed an augmentation in CPITN scores with an increase in saliva TMAO (per 10 units, β = 1.07, *p* = 0.01), and indoxyl sulfate (per 10 units, β = 1.26, *p* = 0.04), meaning worse periodontal health.

## 3. Discussion

We describe, for the first time, the total concentrations of the eight uremic toxins and three precursors in the saliva and their association with dental health, the alveolar bone, and the periodontium, in the CKD population. 

An increase in the DMFT score leads to a reduction in the FST score, due to teeth loss. Teeth degradation is related to a progressive destruction of tooth components (enamel, dentin, cementum) by pathogenic oral microbiota. It was shown that during the process of dental tissues alteration (chronic apical periodontitis, periapical abscess, pulpitis), an inflammatory process occurred, and several oxidative stress markers were detected in the dental pulp and in saliva [4].

More recently, among 56 inflammatory markers searched in children’s saliva, only CSF1 was positively correlated with the presence of caries [5].

In our study, the association between DMFT/FST scores and the saliva levels of hippuric acid or indoxyl sulfate was observed for the first time.

Overall, these observations emphasise the role of dental caries and subsequent tooth alteration in the development of local inflammation that can be detected in saliva samples.

The alveolar bone is the specific area where teeth are implanted in the maxilla and mandible. In CKD patients with an elevation of PTH, a switch has been reported from the trabecular alveolar bone to a granular bone [6]. Data from the literature show that hippuric acid inhibits osteoclastogenesis in vitro [7]. Indoxyl sulfate was reported to be involved in renal osteodystrophy through the inhibition of the differentiation and maturation of osteoclast precursors and the induction of osteoblast apoptosis [8]. Periodontitis is a chronic disease characterised by a loss of periodontal attachment (alveolar bone + periodontal ligament + tooth cementum), related to a chronic local inflammation [9]. This lesion is the consequence of unbalance between patient inflammatory reactivity and its gingival microbiome. Some bacteria crucial to periodontitis development are more prevalent in the CKD population [10]. But the influence of CKD on periodontitis genesis is more complicated, with an interplay between periodontal tissue and CKD and some common mechanisms based on inflammatory state, impaired immune clearance, and other factors, such as xerostomia and the modification of the saliva pH [11]. Saliva composition or flux appeared as a major factor in the development of oral biofilm and, therefore, in the emergence of gingivitis and/or periodontitis [12].

The CPITN score is the direct representation of the quality and integrity of the periodontium. We observed, in our study, a correlation between CPITN and TMAO or indoxyl sulfate in saliva. TMAO and indoxyl sulfate are well-described uremic toxins, correlated with the progression of chronic kidney disease and/or vascular disease [13,14].

TMAO is involved in inflammatory response with endothelial hyperpermeability and reactive oxygen species production that may, importantly, be involved in the development of periodontal lesions. The effects of indoxyl sulfate on endothelial cells are reported with similar impacts to TMAO. The production of uremic toxins, such as indoxyl sulfate and TMAO, is directly linked to the diversity and composition of gut microbiota [15]. Differences in terms of the gut environment have been reported between end-stage renal disease patients and patients with normal renal function [16,17] Thus, we may hypothesise that these two uremic toxins are involved in the genesis of periodontal lesions through the oral and gut microbiota. These hypotheses may be confirmed via further experiments on saliva bacterial ecology but also on the impact of diet modification on the level of saliva uremic toxins and the development of oral lesions [18].

The major strengths of this study include the first-time evaluation of CKD dental health abnormalities, and the evaluation of the relationship between saliva uremic toxins and these kidney disease dental health abnormalities. Our study also has limitations, which include the small size of the study, the log transformation of some variables for analysis, the lack of evaluation of the oral/saliva microbiota, and the exploration of the role of each uremic toxin on CKD dental health alterations. Further experiments are needed to confirm and explore these points.

## 4. Conclusions

Herein, we report the association between various dental scores and saliva uremic toxins and precursors for the first time. These toxins might be implicated in dental health through direct (e.g., involvement in periodontal lesions) or indirect (e.g., inflammation or impact on bone properties) effects, as well as in the pathogenesis of CKD-associated anorexia. These hypotheses remain to be confirmed through further prospective interventional studies.

## 5. Methods

Nephrodent is a one-year prospective study (Clinical Trials NCT02114281) focused on the dental health of patients with non-dialysis CKD or kidney transplant recipients.

All participants were recruited in the Nephrology Unit of the University Hospital of Bordeaux (France). Eligible participants were patients older than 18 years, with no dental care within the past 6 months (recent dental care represented bias of interpretation) and treated in the unit for non-dialysis CKD stages 2 to 5. Pregnant women, patients with a current law affair or unable to make informed choices on their own, and those suffering from psychosis or severe mental impairment were excluded from this study. The protocol was approved by the Consultative Committee for the Protection of Persons participating in December 2013 (N° 2013-A01548-37). All participants provided free, informed, and written consent.

The evaluation of the kidney function was part of the usual care in the Nephology Unit. The glomerular filtration rate (GFR) was used to assess renal function using the Modification of Diet in Renal Disease (MDRD) equation and to categorise patients in each stage of CKD according to the criteria proposed by the Kidney Disease Outcomes Quality Initiative of the National Kidney Foundation (National Kidney Foundation 2002): stage 2 = GFR 60–89 mL/min/1.73 m^2^; stage 3 = GFR 30–59 mL/min/1.73 m^2^, stage 4 = GFR 15–29 mL/min/1.73 m^2^, and stage 5 = GFR < 15 mL/min/1.73 m^2^.

Medical history and sociodemographic variables (sex, age) were collected from medical records. Height and weight (used to calculate body mass index (BMI) in kg/m^2^) were systematically collected at the first visit by the nephrologist. Current and past consumption of alcohol and tobacco (current or ex-smoker) were also recorded by nephrologists.

The dental scores evaluated were: −the “Decayed Missing Filled per Tooth” (DMFT) score [19], which reflected the number of teeth decayed, missing, or filled;−the “Filled and Sound Teeth” (FST) score [20], which quantified the number of functional teeth, the number of functional tooth units, and the number of teeth to be extracted; −the “Community Periodontal Index of Treatment Needs” (CPITN) [21], which determined the periodontal health;−the “Simplified Oral Hygiene Index” (OHIS). The level of hygiene was measured using OHIS, a scale (range: 0–6) that quantifies dental plaque and calculus on six index teeth [22].

The DMFT and FST scores are related on the teeth, whereas CPITN and OHIS are focused on periodontium and dental hygiene, respectively; the overall results show that all the dental criteria evaluated were altered compared to reference values. 

Total concentrations of 8 uremic toxins (trimethylamine N-oxide -TMAO-, Indoxyl Sulfate, P-cresyl-sulfate, Indole 3-acetic acid, 3-carboxy-4-methyl-5-propyl-2-furanpropanoic acid -CMPF-, Kynurenine, Hippuric acid, and Phenylacetylglutamine) and 3 precursors (Tyrosine, Phenylalanine, and Tryptophan) in the saliva were evaluated via the method of liquid chromatography coupled to tandem mass spectrometry (LC-MS/MS) [23]. 

In the Nephrodent study, saliva samples were immediately frozen after the visit and stored at −80 °C until the day of analysis. The saliva sample preparation and protocol were reported by Fabrese et al. [23]. Analyses were performed in Inserm U1173 unit with the same method and data acquisition.

The LC-MS/MS method for saliva samples was validated by Fabresse et al. with saliva samples from healthy patients. The work demonstrated a correlation between plasma and saliva samples [23].

We first tested the univariable associations between saliva uremic toxins and precursors, and DMFT, FST (log-transformed), CPITN, and OHIS (log-transformed) scores. Only those associations with a *p*-value ≤ 0.2 were further investigated in multivariable analyses. Given the limited sample size of our study, and in order to preserve the statistical power, we wanted to limit the number of variables to be included in our multivariable analyses. Using a threshold univariable *p*-value of 0.2 is a current method to do so. The selected associations between saliva uremic toxins and DMFT, FST, and OHIS scores used multivariable linear regression models. The selected associations between saliva uremic toxins and the CPITN score used a multivariable ordinal logistic regression model. All models were adjusted for gender, age, BMI, alcohol consumption, tobacco smoking, and CKD stage.

## Figures and Tables

**Table 1 toxins-15-00666-t001:** Levels of uremic toxins and precursors detected in the saliva of CKD patients (Nephrodent Study).

**Toxins**	**N**	**Mean (Standard Deviation) (ng/mL)**	**[Min–Max]**
TMAO	88	32.32 (108.36)	[0.00–806.36]
Tyrosine	88	8974.61 (6692.03)	[1562.00–29,182.69]
Phenylalanine	88	5662.19 (4624.23)	[739.40–24,754.94]
Tryptophan	88	403.26 (706.18)	[25.00–5191.90]
Indoxyl Sulfate	88	23.14 (30.21)	[0.00–186.60]
P-cresyl-sulfate	88	75.34 (445.77)	[0.00–3980.77]
Indole 3-acetic acid	88	297.01 (584.11)	[2.50–3751.99]
CMPF	88	6.76 (11.49)	[2.81–102.41]
**Toxins**	**N**	**N (%)**	**[min–max]**
Kynurenine	88	undetectable 7 (8.64%) dectable 74 (91.36%)	
Hippuric acid	88	undetectable 61 (75.31%) detectable 20 (24.69%)	
Phenylacetylglutamine	88	undetectable 69 (85.19%) detectable 12 (14.81%)	

**Table 2 toxins-15-00666-t002:** Multivariate analysis of the association between dental scores and the levels of uremic toxins detected in the saliva of CKD patients (Nephrodent Study).

Saliva Uremic Toxins	DMFT Score (n = 78)			log(28—FST Score + 1) * (n = 78)			Log(OHIS Score + 1) (n = 69)			CPITN Score		
	$\hat{\beta }$	95% CI	*p*-Value	$\hat{\beta }$	95% CI	*p*-Value	$\hat{\beta }$	95% CI	*p*-Value	$\hat{OR}$	95% CI	*p*-Value
For an increase of 10 units of TMAO				0.02	[−0.00; 0.04]	0.05				1.07	[1.02; 1.14]	0.01
For an increase of 100 units of Tyrosine	0.018	[−0.004; 0.04]	0.11									
For an increase of 100 units of Phenylalanine	0.03	[−0.007; 0.06]	0.11									
Presence of Hippuric acid	3.28	[−0.33; 6.90]	0.07	0.57	[0.10; 1.04]	0.018						
Presence of Phenylacetylglutamine	3.43	[−0.97; 7.83]	0.12	0.54	[−0.03; 1.12]	0.06	−0.29	[−0.57; −0.02]	0.03			
For an increase of 10 units of Indoxyl sulfate				0.06	[−0.019; 0.13]	0.14				1.26	[1.00; 1.58]	0.04
For an increase of 100 units of p-cresyl-sulfate	0.30	[−0.03; 0.64]	0.08				−0.05	[−0.17; 0.07]	0.11	1.60	[0.82; 3.10]	0.17
For an increase of 100 units of CMPF	1.24	[−0.09; 2.57]	0.07									
For an increase of 100 units of Indol-3-acetic acid				0.02	[−0.01; 0.06]	0.24						

Multivariable analyses adjusted for gender, age, body mass index, alcohol consumption, tobacco smoking, and renal disease stage. * The FST score transformation inverts the direction of the association; a positive $\hat{\beta }$ means that the FST score decreases for an increase in the saliva uremic toxins. The beta values are the coefficients of linear regressions. They represent the mean increase in dental scores for an increase in the quantitative explanatory variables, or the mean difference in dental scores between the categories of qualitative variables.

## Data Availability

The data presented in this study are available on request from the corresponding author.

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
