# Peer review of "Association between Dental Scores and Saliva Uremic Toxins"

_toxins, 2023, doi:10.3390/toxins15110666_

Round 1
Reviewer 1 Report
Comments and Suggestions for Authors
I have reviewed the manuscript toxins-2587115. The manuscript fills a critical research gap by exploring the association between dental scores and saliva uremic toxins in CKD patients. Although robust and informative, the paper requires revisions across all sections to improve clarity, depth, and academic rigor.
Methods Section
1. Inclusion and Exclusion Criteria
- Please clarify the rationale for excluding patients with recent dental care. Understanding this will add depth to this article's methodology.
2. Statistical Methods
- Consider including a dedicated section that details the statistical approaches used for data analysis.
3. Analytical Methods (LC-MS/MS)
- A plain language summary of the LC-MS/MS methods would make this section more accessible to a broader audience.
4. References
- Ensure that all citations are formatted consistently throughout the paper.
5. Statistical Analysis
- While using univariable and multivariable models is well-justified, the rationale for selecting a p-value threshold of ≤0.2 for inclusion in multivariable analyses needs clarification.
6. Abbreviations
- All abbreviations such as DMFT, FST, CPITN, OHIS, and TMAO should be defined upon their first usage for clarity.
Results Section
7. Uremic Toxins and Beta Coefficients
- Please include the concentrations of the uremic toxins and elaborate on the clinical significance of the beta coefficients.
8. CPITN Score
- Clarify what "improvement of CPITN score means," especially since a higher score usually indicates worse periodontal health.
9. CKD Stages
- It would be valuable to discuss whether the results are stage-specific or if they hold across various CKD stages.
Discussion Section
10. Limitations and Future Research
- An explicit discussion of the study's limitations will add depth to the paper. Suggestions for future research based on identified gaps would be beneficial.
11. Impact on CKD Management
- The paper mentions potential dietary and gut microbiota impacts on oral health. Please elaborate on how these findings could influence CKD management strategies.
Conclusion Section
12. Terminology
- In the conclusion, clearly define "direct or indirect effects" to ensure clarity.
Minor Comments
13. Grammar and Syntax
- Some minor grammatical errors need rectification for a more polished manuscript, e.g., "excluded of this study" should be corrected to "excluded from this study."
14. List of Abbreviations
- Consider providing a list of abbreviations like DMFT, FST, and CPITN at the beginning or end of the paper for easy reference.
Author Response
Please, find here the revised version of our paper entitled: “Association between dental scores and saliva uremic toxin”. We thank the reviewer for his comments, we believe that they have helped us to improve our manuscript. Our answers to the different points are listed below.
Reviewer 1
The manuscript fills a critical research gap by exploring the association between dental scores and saliva uremic toxins in CKD patients. Although robust and informative, the paper requires revisions across all sections to improve clarity, depth, and academic rigor.
- Please clarify the rationale for excluding patients with recent dental care. Understanding this will add depth to this article's methodology.
Exclusion of patients with recent dental care was based on the fact that we have excluded all bias in the interpretation of our results. Recent dental care may directly influence dental parameters in a patient compared to another patient (without dental care) at the same stage renal disease.
- Consider including a dedicated section that details the statistical approaches used for data analysis.
Statistical approaches were described in result section. We now include it in the statistical method dedicated section (Methods).
- A plain language summary of the LC-MS/MS methods would make this section more accessible to a broader audience.
We added clarification in the main text to improve the accessibility to the audience, lines 37-43.
- Ensure that all citations are formatted consistently throughout the paper.
We thank the reviewer for the comment and checked all the citations.
- While using univariable and multivariable models is well-justified, the rationale for selecting a p-value threshold of ≤0.2 for inclusion in multivariable analyses needs clarification.
Given the limited sample size of our study, and in order to preserve the statistical power, we wanted to limit the number of variables to be included in our multivariable analyses. Using a threshold univariable p-value of 0.2 is a current method to do so. We included the justification in the main text, lines 241-243.
- All abbreviations such as DMFT, FST, CPITN, OHIS, and TMAO should be defined upon their first usage for clarity.
We defined all the abbreviations.
- Please include the concentrations of the uremic toxins and elaborate on the clinical significance of the beta coefficients.
We modified Table 1. The beta are the coefficients of linear regressions. They represent the mean increase in dental scores for an increase in the quantitative explanatory variables, or the mean difference in dental score between the categories of qualitative variables. We added clarification concerning beta coefficient in the main text, lines 102-104.
- Clarify what "improvement of CPITN score means," especially since a higher score usually indicates worse periodontal health.
Increase of CPITN scores associated to the augmentation of TMAO or indoxyl sulfate means that patients need more dental care/treatment linked to worse periodontal health (lines 115-116).
- It would be valuable to discuss whether the results are stage-specific or if they hold across various CKD stages.
As all our multivariable analyses were adjusted for CKD stages, our results hold across CKD stages.
- An explicit discussion of the study's limitations will add depth to the paper. Suggestions for future research based on identified gaps would be beneficial.
Major strengths of this study include the first time evaluation of CKD dental health abnormalities, and the evaluation of the relationship between saliva uremic toxins and these kidney disease dental health abnormalities. Our study also has limitations, which include the small size of the study, the lack of evaluation of oral/saliva microbiota, and the exploration of the role of each uremic toxin on CKD dental health alterations. Further experiments are needed to confirm and explore these points.
We added these points at the end of the discussion, line 164 to169.
- The paper mentions potential dietary and gut microbiota impacts on oral health. Please elaborate on how these findings could influence CKD management strategies.
Depending on the type of dental and/or saliva microbiota, we may adapt the dietary of each patient to modulate the proportion of each bacteria population and then modified the production of uremic toxins, level of inflammation.
- In the conclusion, clearly define "direct or indirect effects" to ensure clarity.
We clarified the direct and indirect effects of uremic toxins (lines 173-174).
- Some minor grammatical errors need rectification for a more polished manuscript, e.g., "excluded of this study" should be corrected to "excluded from this study."
We corrected the sentence.

Reviewer 2 Report
Comments and Suggestions for Authors
Brief observational study testing correlations between putative uremic retention solutes and dental health scores
Paper lacks value for several reasons
- correlations are weak and few of them
- explanatory factors that drive both poor dentition and renal disease are not explored , such as poverty and diet
- disturbing that uremic retention solutes are not correlated with CKD stage
- supplementary tables are carelessly presented - for example, they are tables of numbers, but there is no statement of what the numbers are! beta coefficients? p values?
Comments on the Quality of English Languagemultiple minor errors require editing
Author Response
Please, find here the revised version of our paper entitled: “Association between dental scores and saliva uremic toxin”. We thank the reviewer for his comments, we believe that they have helped us to improve our manuscript. Our answers to the different points are listed below.
Reviewer 2
Brief observational study testing correlations between putative uremic retention solutes and dental health scores
Paper lacks value for several reasons:
- correlations are weak and few of them
- -explanatory factors that drive both poor dentition and renal disease are not explored, such as poverty and diet
We thank the Reviewer, but diet and socioeconomic categories were not associated to dental status. All our multivariable analyses were adjusted for gender, age, Body Mass Index, alcohol consumption, tobacco smoking, and renal disease stage (lines 99-100).
- disturbing that uremic retention solutes are not correlated with CKD stage
We are agreed with the reviewer on this point. However due to the limited number of patients included in the present preliminary study, we are not able to take this action. Additional analysis on uremic toxins and oral bacterial representation will be performed to answer the stage-specific effects (lines164-169).
- supplementary tables are carelessly presented - for example, they are tables of numbers, but there is no statement of what the numbers are! beta coefficients? p values?
As asked by the reviewer, we clarified the supplementary tables. Only p-values are presented in supplementary tables.
We sincerely thank all reviewers and the editor for their constructive criticisms. We hope to have properly answered the points raised by the reviewers and that the revised manuscript is now suitable for publication in Toxins.
Yours faithfully,

Round 2
Reviewer 2 Report
Comments and Suggestions for Authors
MS much improved - thank you to the authors.
- the FST score is poorly described. The rationale for log transformation and (1-log(FST) needs explanation in much more detail. If I undetand correctly, DMFT is roughly "bad teeth" and FST is roughly "good teeth" but this needs a lot more detail
- Table 1 is much more helpful but still confusing. It looks as thoguht the authors are trying to test te impact of the uremic solute on dental score, while controlling for precursor concentration? Is this so? There might be a tidier way of showing this.
- hsitograms of the scores mighr be a good figure to add
Author Response
Please, find here the revised version of our paper entitled: “Association between dental scores and saliva uremic toxin”. We thank the reviewer for his comments. Our answers to the different points are listed below.
Reviewer 2
- the FST score is poorly described. The rationale for log transformation and (1-log(FST) needs explanation in much more detail. If I understand correctly, DMFT is roughly "bad teeth" and FST is roughly "good teeth" but this needs a lot more detail.
We thank the reviewer. These informations are provided in the method section.
- Table 1 is much more helpful but still confusing. It looks as thought the authors are trying to test te impact of the uremic solute on dental score, while controlling for precursor concentration? Is this so? There might be a tidier way of showing this.
We performed a description study of uremic toxins and dental score and secondly analysed the correlation. We didn’t perform any intervention.
- histograms of the scores might be a good figure to add
We added histograms in supplemental informations.
We sincerely thank the reviewer for his constructive criticisms. We hope to have properly answered the points raised and that the revised manuscript is now suitable for publication in Toxins.
Yours faithfully,

Round 3
Reviewer 2 Report
Comments and Suggestions for Authors
I thank the authors for adding the information. My recommendation is to accept with minor revision. I suggest but do not insist that the central tendency and dispersion of the variables be presented as median and IQR, since based on what is presented (and that the authors log-transformed them for analysis, they are probably not normally distributed
Its not clear that normal amino acids need to be described as toxins
not clear why some toxins presented as detected/not detected rather than more quantitative statistics- maybe I missed why.
Comments on the Quality of English Language
ok
Author Response
Please, find the comments about our paper entitled: “Association between dental scores and saliva uremic toxin”. We thank the reviewer for his comments. Our answers to the different points are listed below.
Reviewer 2:
My recommendation is to accept with minor revision. I suggest but do not insist that the central tendency and dispersion of the variables be presented as median and IQR, since based on what is presented (and that the authors log-transformed them for analysis, they are probably not normally distributed).
We added a comment in limitation paragraph.
Its not clear that normal amino acids need to be described as toxins.
We modified this point in the main text.
not clear why some toxins presented as detected/not detected rather than more quantitative statistics- maybe I missed why.
The sensitivity of our method did not allow us to detect the very slow concentrations of certain UTS
We hope to have properly answered the points raised and that the revised manuscript is now suitable for publication in Toxins.
Yours faithfully,
